# Optical and Scanning Electron Microscopy Thrombus Findings in Patients with STEMI Undergoing Primary Versus Rescue PCI

**DOI:** 10.3390/biomedicines13092235

**Published:** 2025-09-11

**Authors:** Stella Marinelli Pedrini, Thiago P. A. Aloia, André H. Aguillera, Paula M. P. S. Gomes, Jamil R. Cade, Francisco Sandro Menezes-Rodrigues, Bárbara P. Freitas, Marco T. Souza, Francisco A. H. Fonseca, Marcos Danillo Oliveira, Breno O. Almeida, Andrey J. Serra, Renato D. Lopes, Rita Sinigaglia-Coimbra, Adriano Caixeta

**Affiliations:** 1Faculdade Israelita de Ciências da Saúde Albert Einstein, São Paulo 05652-900, Brazil; stella.mpmp@gmail.com (S.M.P.); paulagomes130997@gmail.com (P.M.P.S.G.); 2Department of Interventional Cardiology, Hospital Israelita Albert Einstein, São Paulo 05652-900, Brazil; thiagoaloia1@hotmail.com (T.P.A.A.); barbara.freitas@einstein.br (B.P.F.); breno.almeida@einstein.br (B.O.A.); 3Centro de Microscopia Eletrônica (CEME), Universidade Federal de São Paulo, São Paulo 04039-032, Brazilrita.sinigaglia@unifesp.br (R.S.-C.); 4Hospital Santa Marcelina, São Paulo 08270-070, Brazil; jamilcade@me.com; 5Department of Medicine, Discipline of Cardiology, Escola Paulista de Medicina, Universidade Federal de São Paulo, Av. Napoleão de Barros, 714 São Paulo, São Paulo 05652-000, Brazil; sandro.rodrigues@unifesp.br (F.S.M.-R.); marcotuliomed@hotmail.com (M.T.S.); fahfonseca@terra.com.br (F.A.H.F.); mdmarcosdanillo@gmail.com (M.D.O.); andreyserra@gmail.com (A.J.S.); 6Postgraduate Program of Interdisciplinary Science Surgery, Universidade Federal de São Paulo, São Paulo 05652-000, Brazil; 7Duke Clinical Research Institute, Duke University Medical Center, Durham, NC 27710, USA; renato.lopes@duke.edu

**Keywords:** acute myocardial infarction, percutaneous coronary intervention, fibrinolytic failure, thrombus, optical microscopy, scanning electron microscopy

## Abstract

**Background**: The mechanisms underlying fibrinolysis failure in patients with STEMI who are undergoing a pharmacoinvasive strategy appear to be multifactorial and may be associated with the thrombus’s architecture and composition. **Objective**: We aimed to compare the thrombus composition in patients with STEMI who were undergoing rescue percutaneous coronary intervention (rPCI) versus primary PCI (pPCI) using optical microscopy (OM) and scanning electron microscopy (SEM). Methods: Fifty-three patients were prospectively enrolled, with twenty-five undergoing rPCI and twenty-eight undergoing pPCI. After thrombus aspiration, each harvested fragment was divided into two pieces: one was analyzed using OM with a 60× magnifying lens on hematoxylin–eosin-stained samples, and the other with SEM at 5000× magnification. **Results**: Patients who underwent rPCI had significantly higher C-reactive protein levels and a longer ischemic interval at admission compared to those treated with pPCI (9.92 h [range: 1.58–106.17] vs. 2.14 h [range: 0–48]; *p* < 0.001). Optical microscopy analysis revealed that thrombi from rPCI patients exhibited a significantly higher erythrocyte area percentage (18.36% [range: 0.3–50.08] vs. 0.91% [range: 0–70.1]; *p* = 0.001), a lower fibrin content as assessed by optical microscopy (79.49% [range: 49.2–98.25] vs. 94.43% [range: 29.19–99.92]; *p* = 0.006), and a greater amount of cholesterol crystals as measured by SEM (1.73 μm^2^ [range: 0–18.51] vs. 0.08 μm^2^ [range: 0–0.71]; *p* < 0.001). **Conclusions**: The thrombus composition of patients with STEMI who are undergoing rPCI had higher amounts of erythrocytes and cholesterol crystals and a lesser area occupied by fibrin compared to those undergoing pPCI. The composition of thrombi in rPCI could potentially contribute to the failure of fibrinolytic therapy within a pharmacoinvasive strategy.

## 1. Introduction

Primary percutaneous coronary intervention (pPCI) is the preferred reperfusion strategy for patients with ST-elevation myocardial infarction (STEMI) when performed within 120 min [1,2]. However, the timely delivery of pPCI is often impeded by logistical challenges and systemic delays, which are particularly prevalent in developing nations. As a result, many patients with STEMI initially present to hospitals without PCI capabilities, necessitating fibrinolysis therapy before transfer to a PCI-enabled facility. On the one hand, a pharmacoinvasive approach—early fibrinolysis followed by timely PCI—provides a viable alternative reperfusion strategy. On the other hand, fibrinolysis frequently fails in patients with STEMI and has been associated with increased morbidity and mortality [3]. For instance, in the STREAM trial [4], one-third of the subjects did not achieve reperfusion with this pharmacological strategy, despite being treated within 3 h of symptom onset.

The mechanisms underlying the limited efficacy of fibrinolytic agents in STEMI treatment are not yet fully understood. Several factors may contribute to this limitation, including a pro-coagulant state [5,6], the paradoxical effects of fibrinolysis—such as platelet activation [7,8]—the ischemic time [9], patient demographics, and the intrinsic composition and architecture of the thrombus [10]. Contemporary microscopic pathological classifications challenge the traditional macroscopic characterization of arterial thrombi as ‘red’ or ‘white’ [11], suggesting instead that thrombi exhibit heterogeneous compositions. It has been proposed that thrombi be classified into two categories: ‘recent,’ characterized by an intact but disorganized structure, and ‘antique,’ marked by necrosis, apoptosis, leukocyte infiltration, connective tissue deposition, and smooth muscle cell involvement, which is indicative of a more organized structure [12,13,14]. Histological evaluation of a thrombus’ features during STEMI provides a valuable opportunity [15] to perform a detailed examination of its composition, dynamic formation, and architectural structure. Accordingly, our study aimed to investigate the histopathological differences in thrombus composition, using both optical microscopy (OM) and scanning electron microscopy (SEM), between patients treated with rescue (rPCI) and those who underwent pPCI for STEMI.

## 2. Methods

### 2.1. Study Design

This prospective, non-randomized, multicenter study was conducted across three hospitals in São Paulo, Brazil: Hospital Israelita Albert Einstein, Escola Paulista de Medicina at Universidade Federal de São Paulo, and Hospital Santa Marcelina. Between 2015 and 2017, 53 patients with STEMI were prospectively enrolled. Of these, 28 underwent pPCI as a possible control group, while 25 underwent rescue rPCI. The sample size was arbitrarily determined based on a study by Silvain et al. [10], which also evaluated coronary thrombus composition in patients with STEMI. The decision to perform pPCI or rPCI followed institutional protocols and resource availability, with pPCI, the gold standard, being prioritized whenever feasible. All patients in the rPCI group received tenecteplase (30–50 mg bolus, adjusted for weight) and underwent rPCI as early as possible after documented fibrinolytic failure. Following the tenecteplase bolus, patients were given a loading dose of clopidogrel and enoxaparin before transport to a PCI-capable hospital. Additionally, 100–200 mg of aspirin was administered in the emergency department for immediate effect, followed by a daily oral dose of 100 mg during hospitalization. Failed fibrinolysis was defined as <50% ST-segment resolution in the lead with the highest segment elevation on a follow-up electrocardiogram performed 60–90 min after fibrinolysis, combined with TIMI flow 0–1 observed on angiography [16]. Consequently, all patients classified as rPCI were treated within the public health system, where pPCI was not immediately available and was typically performed only after fibrinolysis failure. The study protocol was approved by the local ethics committee (Hospital Israelite Albert Einstein, CAAE: 20940514.3.2002.0071) in accordance with the Declaration of Helsinki. Written informed consent was obtained from all participants prior to their inclusion in this study.

### 2.2. Thrombus Harvesting

Patients with STEMI who were eligible for this study underwent coronary angiography of the culprit lesion. After crossing the lesion with a guidewire but before balloon dilation or stent implantation, a manual catheter (Export, Medtronic, Santa Rosa, CA, USA; or Pronto, Vascular Solutions, Inc., Minneapolis, MN, USA) was used for thrombectomy. The harvested samples were rinsed with saline solution and then divided into two parts, with each part being processed for either OM or SEM assessment. Once the material was prepared, a detailed analysis was conducted to quantify the area occupied by thrombus components, including erythrocytes, leukocytes, fibrin, platelets, and cholesterol crystals. Notably, cholesterol crystals and platelets were assessed only via SEM, as OM using hematoxylin–eosin staining does not allow for their visualization.

### 2.3. Optical Microscopy

Each sample was fixed in 10% buffered formalin for 24 h and then embedded in paraffin. Up to four sequential 5 μm thick histological sections were cut using a rotary microtome and stained with hematoxylin and eosin (H&E). For analysis, three to four images (depending on thrombus size) were captured from randomly selected fields. To minimize bias and ensure quality, slides were first examined at 10× magnification to identify artifact-free regions with clear histological features (e.g., free from tissue folds or staining inconsistencies). These fields were then randomly selected within those regions for imaging at 60× magnification. Semi-automated structural analysis was performed using cellSens Dimension V1 software (Olympus Corporation, Tokyo, Japan), which segmented objects by color thresholding based on H&E staining (Figure 1). Red blood cells, white blood cells, and fibrin were identified as distinct, non-connected objects. Threshold values were adjusted to optimize pixel selection according to staining intensity. Measurement parameters, including object class, count, and summed area per component, were displayed in the results sheet, which enabled direct comparison of components by their assigned colors. Manual adjustments were made when color segmentation did not accurately represent structures in the original H&E images.

### 2.4. Scanning Electron Microscopy (SEM)

The material was fixed in 2.5% glutaraldehyde buffered with 0.1 M sodium cacodylate. The processing procedure followed the GOTO protocol [17], which involves sequential washing with tannic acid and osmium tetroxide, critical point drying, and gold sputter-coating. The morphometric analysis methods were specifically developed for this study, drawing on the expertise of Silvain et al. [10], as no established protocol exists in the literature that is tailored to this type of material or this study’s objectives. The established method involved randomly selecting 6 images captured at 5000× magnification from an average of 30 images per sample. A 190 µm^2^ grid with a random offset was then applied to each selected image. Morphometric grid-based methods typically aim for the grid cell size to accommodate the visualization of 3 to 4 units of the studied structure. Consequently, the grid size was determined based on the average dimensions of leukocytes and erythrocytes. This parameter of a 190 µm^2^ grid was consistently applied to other structures—fibrin, platelets, and cholesterol crystals—to ensure uniformity in the appraisal process. Using the same grid and offset for all structures enhanced the representativeness of the material’s composition and organization. Since these analyses are highly labor-intensive, area measurements were conducted in alternating full grid spaces, starting with the first complete square at the top left of the image. For each image, 5 or 6 grid spaces were quantified, depending on the random offset, which resulted in a minimum analyzed area of 5700 µm^2^ per patient. This area corresponded to approximately 50% of the total thrombus surface. Although the analyzed areas represent only a portion of the total surfaces of the thrombi, random field selection and systematic sampling were employed to enhance representativeness. Measurements were performed by manually contouring each structure using the tools available in ImageJ 1.50i (National Institutes of Health, Bethesda, MD, USA). Structures that were unclear or could not be confidently identified were categorized as “other” to preserve the total absolute area and ensure accurate percentage calculations.

### 2.5. Statistical Analysis

Categorical variables were compared using the χ^2^ test or Fisher’s exact test, as appropriate. Continuous variables were analyzed with Student’s *t*-test or the Mann–Whitney U test and are expressed as either median (range) or mean ± SD, with normality assessed using the Kolmogorov–Smirnov test. The OM and SEM analyses were not intended to yield identical results, as the methods are complementary: OM focuses on the thrombus core, whereas SEM examines its surface. Correlations between the ischemic interval and thrombus composition were assessed using Spearman’s rank test, given the potential confounding effect of the ischemic interval on thrombus characteristics. Statistical analysis was performed using SPSS Statistics 22.0 (IBM, Armonk, NY, New York), and a *p*-value of <0.05 was considered statistically significant.

## 3. Results

Of the 53 patients included in this study, 42 had sufficient material for both OM and SEM analyses. In five cases, the amount of aspirated thrombus material was insufficient for SEM processing, while, in six instances, OM could not be performed due to a limited sample size.

The demographic characteristics between the groups were generally well balanced, with the exception that patients undergoing rPCI had significantly higher levels of C-reactive protein, total cholesterol, and low-density lipoprotein (LDL) cholesterol. Additionally, the patients undergoing rPCI experienced a longer ischemic interval at admission compared to those treated with pPCI (9.9 h [range: 1.6–106.2] vs. 2.1 h [range: 0–48]; *p* < 0.001). (Table 1).

### 3.1. Histopathological Analysis

All measurements obtained via OM (erythrocytes, leukocytes, and fibrin) and SEM (erythrocytes, leukocytes, fibrin, platelets, and cholesterol crystals) were recorded and expressed as either relative or absolute areas. No significant differences were observed in the absolute areas between the groups. The data were then statistically analyzed to compare the thrombus compositions of the rPCI and pPCI groups. Compared to the patients undergoing pPCI, those in the rPCI group exhibited thrombi with a significantly higher erythrocyte area percentage (18.36% [range: 0.3–50.08] vs. 0.91% [range: 0–70.1]; *p* = 0.001) and a lower fibrin content (79.49% [range: 49.2–98.25] vs. 94.43% [range: 29.19–99.92]; *p* = 0.006), as assessed by optical microscopy (Figure 2). Scanning electron microscopy revealed a significantly greater amount of cholesterol crystals in the rPCI group (1.73 μm^2^ [range: 0–18.51] vs. 0.08 μm^2^ [range: 0–0.71]; *p* < 0.001) (Table 2). Additionally, there was a trend toward a higher leukocyte area in the rPCI group based on SEM analysis (1.44 μm^2^ [range: 0.03–20.65] vs. 0.54 μm^2^ [range: 0–21.88]; *p* = 0.07) (Figure 3).

### 3.2. Ischemic Interval Correlation Analysis

Due to potential collinearity between the ischemic interval and fibrinolytic therapy (Table 1)—as both are influenced by the time required to establish treatment failure and reperfusion through rPCI—we conducted a univariate correlation analysis between the thrombus composition and ischemic interval, without distinguishing between the pPCI and rPCI groups (Table 3). The analysis revealed a weak but statistically significant positive correlation between cholesterol crystals and the ischemic interval (r = 0.364, *p* = 0.014), as well as between erythrocytes and the ischemic interval (r = 0.335, *p* = 0.03).

## 4. Discussion

The primary findings of this study are as follows: compared with patients undergoing pPCI, those receiving rPCI had significantly higher levels of C-reactive protein, total cholesterol, and LDL cholesterol, as well as a longer ischemic interval at hospital admission. Histopathological analysis revealed that the thrombi in rPCI patients contained higher amounts of erythrocytes, less fibrin (as assessed by OM), and a greater quantity of cholesterol crystals (as assessed by SEM).

The strength and novelty of our study lie in the dual analysis of the harvested thrombi using both OM and SEM on the same patient and the same thrombus sample.

Various factors may serve as a critical underlying substrate contributing to fibrinolytic failure, including the thrombus composition, which represents a complex interplay of blood components and atherosclerotic plaque constituents [9,10,11,12,13,14]. The composition of thrombi was evaluated by Silvain et al. [10] in 45 patients with STEMI who were undergoing pPCI, and they demonstrated that the ischemic time was positively correlated with the thrombus composition and negatively correlated with the platelet content based on SEM analysis. In the present analysis, when groups were not distinguished, the SEM-based thrombus composition was broadly comparable to the findings of Silvain et al., with similar overall percentages of fibrin (66.14 ± 23.82% vs. 55.9 ± 8.4%), red blood cells (7.38 ± 9.57% vs. 11.5 ± 9%), and platelets (21.61 ± 21.02% vs. 16.8 ± 18%), respectively. However, a negative correlation between the fibrin content and ischemic interval was revealed in the present analysis. These contrasting findings may be attributed to two key factors: (i) fibrinolytic therapy was administered to only 2% of participants in that study, which minimized its influence on thrombus composition, and (ii) a longer ischemic interval was observed in the present analysis (594 min vs. 128 min; or 9.9 h vs. 2.1 h). In the present study, all patients undergoing rPCI received fibrinolytic therapy before thrombus aspiration, which may have contributed to the reduced fibrin content observed in the thrombi. Therefore, the potential influence of fibrinolytic therapy on thrombus composition should be carefully considered when interpreting differences between the rPCI and pPCI groups.

Additionally, the negative correlation between platelets and the ischemic interval reported by Silvain et al. [10] was not observed in our study. Instead, we found a trend suggesting the opposite association, which aligns with fibrinolytic failure. Similarly, Cannon et al. [18] highlighted the detrimental role of platelets in achieving effective reperfusion through fibrinolysis. Platelets play a central role in coronary thrombosis, increasing the risk of complications such as vessel re-occlusion and reinfarction. Consequently, the use of dual antiplatelet therapy with aspirin and P2Y12 inhibitors is critically important following fibrinolytic therapy [19].

A longer ischemic interval has been associated with more organized thrombi, which are often classified as red thrombi and are linked to a higher incidence of clinical complications [20,21]. In our study, the relatively or absolutely higher erythrocyte content observed in rPCI compared to pPCI (as assessed by OM) was expected, given the longer ischemic interval. However, whether this finding is incidental or directly related to fibrinolytic failure remains to be determined. Cholesterol crystals have been implicated in plaque rupture through mechanical injury and inflammation. Abela et al. [22] analyzed 240 aspirated thrombi from patients with STEMI with culprit coronary artery obstruction using SEM, crystallography, and infrared spectroscopy. Moderate to extensive clusters of cholesterol content, confirmed as cholesterol via spectroscopy, were detected in 60% of the patients. Totally occluded arteries demonstrated significantly larger cholesterol crystal clusters compared to partially occluded arteries. The presence of these clusters was also associated with heightened inflammation (elevated interleukin-1β levels), greater arterial narrowing, and impaired reflow after PCI. While the authors did not quantify the percentage of cholesterol crystal occupancy within the thrombi, our findings align with theirs in highlighting the link between cholesterol crystals and an inflammatory response. We observed a positive correlation between admission C-reactive protein levels and cholesterol crystal content, as well as a higher leukocyte count in both OM and SEM analyses. Moreover, inflammatory responses and markers of injury tend to be more pronounced in STEMI cases that involve plaque rupture with core exposure, which includes cholesterol crystals—a factor also associated with worse prognoses [23]. These findings align with our results, suggesting that cholesterol crystals in the plaque or thrombus may contribute to fibrinolytic failure. Plaque rupture, exposing a lipid-rich core with elevated cholesterol content, creates challenges in achieving adequate reperfusion [24], increasing the risk of distal embolization and leading to unfavorable clinical outcomes.

Our findings provide comprehensive and detailed insights into the composition of thrombi in patients with STEMI who are undergoing rPCI compared to those undergoing pPCI. However, larger multicenter studies are needed to corroborate these results and explore the underlying mechanisms driving these differences. Furthermore, it remains to be determined whether research should focus on developing targeted therapeutic interventions based on the composition of thrombi to improve outcomes in patients experiencing fibrinolytic failure.

## 5. Limitations

Although this study was carefully designed, several limitations should be acknowledged. First, the sample size was small, and the pPCI and rPCI groups were not homogeneous. This lack of homogeneity was unavoidable, as patients undergoing rPCI naturally experience longer ischemic intervals compared with those undergoing pPCI. Consequently, the potential collinearity between rPCI and the ischemic interval, and its influence on thrombus composition, cannot be excluded. Additionally, the patients in the rPCI group tended to present with more uncontrolled comorbidities. Second, the limited body of research on this topic poses challenges to standardizing data collection and thrombus analysis methodologies, complicating efforts to ensure consistency in measurement techniques. Third, OM and SEM target distinct areas of the thrombus architecture. OM evaluates the core composition of the thrombus, while SEM predominantly examines surface features. SEM analysis was performed exclusively on the surface of the thrombi, as cross-sectional imaging was not feasible with the applied preparation techniques. These inherent methodological differences may contribute to variations in the quantitative results and should be considered when interpreting the findings. Further, the semi-automated OM measurements could be subject to unintentional observer interference. Fourth, in certain cases, it was not feasible to obtain paired samples for both OM and SEM analysis, which may have introduced potential measurement bias. Moreover, since aspirated thrombi represent only a fraction of the thrombus involved in the atherothrombotic event, they may not comprehensively reflect the composition of the entire thrombus. Fifth, the strong relationship between the ischemic interval and fibrinolytic failure creates challenges for multivariable analysis, making it difficult to isolate individual predictors of fibrinolytic failure. Finally, the present findings should be considered a hypothesis only, and further and larger studies are warranted.

## 6. Conclusions

In patients with STEMI, the thrombus composition differed significantly between those undergoing rPCI and those undergoing pPCI. Specifically, the rPCI thrombi exhibited higher erythrocyte content (by OM), increased cholesterol crystals (by SEM), and a reduced fibrin area (by OM). These findings suggest that thrombi with higher erythrocyte and cholesterol crystal content in patients undergoing rPCI may contribute to the failure of fibrinolytic therapy during a pharmacoinvasive strategy.

## Figures and Tables

**Figure 1 biomedicines-13-02235-f001:**
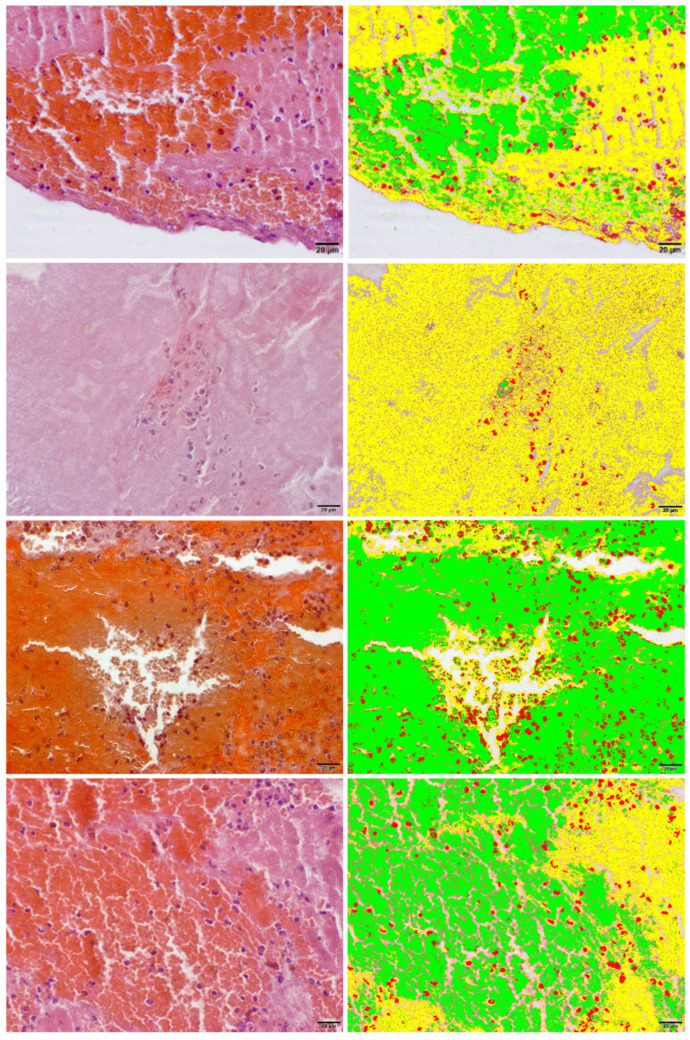
Optical microscopy methodology overview. (**Left**) Column: four hematoxylin and eosin-stained slides before area measurements. (**Right**) Column: the analysis filter was semi-automatically applied based on a color threshold. Upon completion, the software provided an absolute area assessment for each structure (erythrocytes in green, leukocytes in red, and fibrin in yellow).

**Figure 2 biomedicines-13-02235-f002:**
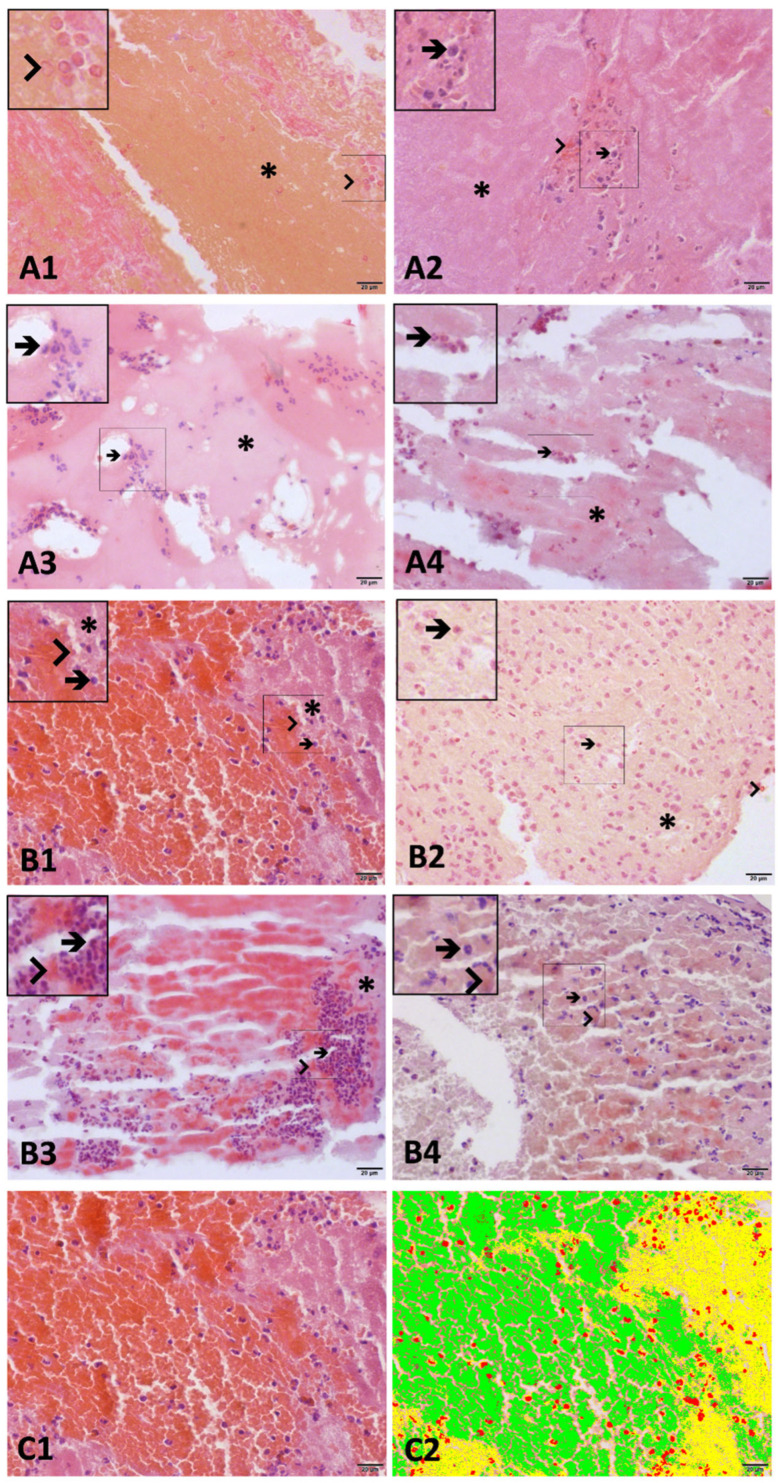
Photomicrographs of thrombi retrieved from patients undergoing primary PCI (**A1**–**A4**) and rescue PCI (**B1**–**B4**), stained with hematoxylin and eosin (H&E). The arrowhead indicates red blood cells, the arrow indicates white blood cells, and the asterisk denotes fibrin. The inset panels in the upper left corner (**A1**–**B4**) display magnified views to facilitate visualization of cellular components. (**C**) Semi-automated structural analysis performed using cellSens Dimension V1 software. Objects were segmented by color thresholding based on H&E staining: erythrocytes are highlighted in green, leukocytes in red, and fibrin in yellow (**C1**,**C2**). Stain: H&E. Scale bar = 20 μm.

**Figure 3 biomedicines-13-02235-f003:**
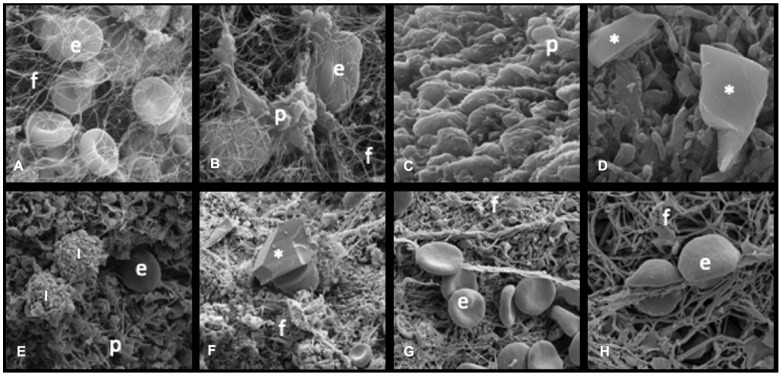
Scanning electron microscopy (SEM) of thrombus composition in patients with myocardial infarction. This series of SEM images provides a detailed examination of thrombus components in patients undergoing rescue PCI (top, (**A**–**D**)) and primary PCI (bottom, (**E**–**H**)). The panels display varying magnifications and sections of the thrombus, highlighting the roles of key cellular and structural elements: Erythrocytes (e): identified by their round, biconcave shapes, erythrocytes are densely packed within the fibrin network, which underscores their significant contribution to thrombus formation. Leukocytes (l): larger than erythrocytes and distinct in their morphology, leukocytes suggest an inflammatory response, being a hallmark of acute coronary events. Platelets (p): smaller and often observed in aggregates, platelets play a pivotal role in thrombus initiation and consolidation. Fibrin (f): composing the fibrous meshwork enveloping the cellular components, fibrin provides essential stabilization and structural integrity to the thrombus, trapping blood cells and reinforcing clot firmness. Cholesterol crystals: identified with an asterisk (*), these crystals are indicative of atherosclerosis, a critical factor in plaque destabilization and subsequent thrombus formation.

**Table 1 biomedicines-13-02235-t001:** Baseline demographic characteristics of patients undergoing pPCI and rPCI.

	Primary PCI (n = 28)	Rescue PCI (n = 25)	Total	*p*-Value
Age, years	50.5 (35–82)	56.5 (38–73)	51.5 (35–82)	0.27
Body mass index, kg/m^2^	28.7 (20.1–38.7)	26.0 (23.3–38.2)	27.2 (20.0–38.7)	0.49
Smoker	46.2%	43.5%	44.9%	0.97
Hypertension	46.2%	65.2%	55.1%	0.18
Dyslipidemia	47.8%	69.6%	58.7%	0.13
Diabetes mellitus	19.2%	30.4%	24.5%	0.36
Family history coronary artery disease	26.9%	34.8%	30.6%	0.55
Congestive heart failure	3.8%	0.0%	2.0%	>0.99
Total cholesterol, mg/dL	169 (89–209)	193 (148–238)	177.5 (89–238)	0.004
Low-density lipoprotein	93.5 (31–131)	122 (66–163)	110 (31–163)	0.004
High-density lipoprotein	42 (23–66)	31.5 (24–75)	35 (23–75)	0.56
Creatinine clearance, mL/min	97.8 (32.9–163.4)	133.5 (45–185)	117.6 (32.9–185)	0.058
Triglycerides, mg/dL	110 (46–332)	105 (59–227)	108 (46–332)	0.37
Creatinine, mg/dL	1.05 (0.58–3.09)	0.82 (0.6–1.62)	0.91 (0.58–3.09)	0.07
C-reactive protein, mg/dL	0.78 (0.04–85)	18.4 (4–120)	1.385 (0.04–120)	0.002
HbA1c, %	5.8 (0.05–15)	5.4 (0.06–8.8)	5.55 (0.05–15)	0.61
Ischemia interval, h	2.1 (0–48)	9.9 (1.6–106.2)	7 (0–106.2)	<0.001

Values are median (range) or % (n).

**Table 2 biomedicines-13-02235-t002:** Histopathological analysis of thrombi from patients undergoing primary versus rescue PCI, according to percentage of area occupancy.

	Primary PCI Mean ± SD	Primary PCI Median (Minimum to Maximum)	Rescue PCI Mean ± SD	Rescue PCI Median (Minimum to Maximum)	Total Mean ± SD	Total Median (Minimum to Maximum)	***p* Value**
**Scanning electron microscopy (%)**							
Erythrocytes	5.94 ± 8.36	2.27 (0–33.88)	8.95 ± 10.71	4.456 (0–39.8)	7.38 ± 9.57	2.81 (0–39.8)	0.19
Leukocytes	2.28 ± 4.72	0.54 (0–21.88)	3.48 ± 4.85	1.44 (0.03–20.65)	2.85 ± 4.77	0.79 (0–21.88)	0.07
Fibrin	73.02 ± 19.48	76.62 (36.4–99.94)	58.66 ± 26.19	65.31 (0.72–96.28)	66.14 ± 23.82	69.64 (0.72–99.94)	0.056
Platelets	18.28 ± 16.84	14.14 (0.04–57.75)	25.22 ± 24.67	16.72 (1.75–99.11)	21.61 ± 21.02	15.43 (0.04–99.11)	0.35
Cholesterol crystals	0.22 ± 0.25	0.08 (0–0.71)	3.42 ± 4.81	1.73 (0–18.51)	1.76 ± 3.67	0.35 (0–18.51)	<0.001
**Optical microscopy (%)**							
Erythrocytes	8.69 ± 16.93	0.91 (0–70.1)	22.36 ± 19.21	18.36 (0.3–50.08)	14.22 ± 18.94	2.26 (0–70.1)	0.001
Leukocytes	2.82 ± 2.55	1.57 (0–9.26)	3.36 ± 2.84	3.88 (0–9.02)	3.04 ± 2.66	2.24 (0–9.26)	0.76
Fibrin	88.49 ± 17.14	94.43 (29.19–99.92)	74.28 ± 18.86	79.49 (49.2–98.25)	82.74 ± 19.01	92.86 (29.19–99.92)	0.006

PCI = percutaneous coronary intervention.

**Table 3 biomedicines-13-02235-t003:** Spearman’s correlation test between thrombus composition and ischemic interval.

Variable	Total		
	Coefficient	n	*p*-Value
**Scanning electron microscopy**			
Erythrocytes	0.051	45	0.74
Leukocytes	0.175	45	0.25
Fibrin	−0.139	45	0.36
Platelets	0.116	45	0.45
Cholesterol crystals	0.364	45	0.014
Others	0.207	45	0.17
**Optical microscopy**			
Leukocytes	0.284	43	0.07
Erythrocytes	0.335	43	0.03
Fibrin	−0.115	43	0.46

## Data Availability

The raw data supporting the conclusions of the article will be made available by the authors on request.

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
