# Peer review of "Optical and Scanning Electron Microscopy Thrombus Findings in Patients with STEMI Undergoing Primary Versus Rescue PCI"

_biomedicines, 2025, doi:10.3390/biomedicines13092235_

Round 1
Reviewer 1 Report
Comments and Suggestions for Authors
This study obtained the thrombus materials from two types of STEMI patients: patients undergoing primary percutaneous coronary intervention (pPCI, n=28) and patients undergoing rescue percutaneous coronary intervention (rPCI, n=25). The harvested thrombus fragment was divided into two pieces. One piece was analyzed by optical microscopy (OM) on hematoxylin–eosin-stained samples, and the other by scanning electron microscopy (SEM), respectively, to obtained the thrombus composition. Results showed thrombi in patients undergoing rPCI exhibited a higher erythrocyte content, more cholesterol crystals, and reduced fibrin area. This is an interesting topic, however, to my opinion, there are some serious issues in the work against its publication:
1) Clinical significance: One main result is that thrombi in patients with rPCI had reduced fibrin area, compared to pPCI. Since the patients with rPCI had Fibrinolytic therapy before harvesting the thrombus materials,
2) Table 2. According to the data presented in Table 2, it is hard to believe that the quantitative analysis of HE images and SEM image is accurate enough. In Table 2, both OM and SEM methods can detect Erthrocytes area. But the results from two methods are dramatically different, and statistical results are also different (one says that pPCI and rPCI are significant different, the other says otherwise). In my opinion, the conclusion is not convincing if it is method-dependent.
3) Section 2.3. The HE images was analyzed using a semi-automated structural analysis based on color thresholds. What are the thresholds to segment different thrombus components, please provide more details.
If the authors used one threshold for all HE images, how did the authors address the color variation during the HE staining process.
4) Does all patients undergoing rPCI have a failed Fibrinolytic therapy?
5) Section 2.4. Each SEM image was analyzed with a minimum analyzed area of 5700 μm² per patient. What is the percent of the analyzed area (5700 μm²) over the entire sample area? If the percent is too small, could the analyzed area be a good/accurate presentative to the whole sample?
6) Results Section. Please provide more details on why 5 thrombi only had OM, and 6 only had SEM? What are the sample sizes of these thrombi? And what is the minimal sample size eligible for OM or SEM in this study?
7) Figure 2, redundant caption in the figure.
Author Response
Please see below the response to the reviewer's comments below and the clear version of the manuscript attached.
Reply to Reviewers
Manuscript Title: Optical and Scanning Electron Microscopy Thrombus Findings in STEMI Patients Undergoing Primary Versus Rescue PCI
Manuscript ID: biomedicines-3717514
Journal: Biomedicines
Dear Editor and Reviewers,
We sincerely thank the Editor and reviewers for their thoughtful and constructive comments on our manuscript. We greatly appreciate the time and effort dedicated to reviewing our work and for the valuable suggestions that have helped us enhance the clarity and scientific rigor of the manuscript. Below, we provide a detailed, point-by-point response to each comment. All changes made in the revised manuscript are highlighted in yellow for easy reference.
Reviewer Comments and Authors’ Responses
Reviewer Comment 1:
Clinical significance: One main result is that thrombi in patients with rPCI had reduced fibrin area compared to pPCI. Since the patients with rPCI had fibrinolytic therapy before harvesting the thrombus materials…
Response:
We thank the reviewer for this important observation. Indeed, all patients in the rPCI group received fibrinolytic therapy (tenecteplase) before thrombus harvesting. We agree that fibrinolysis may have influenced thrombus composition, potentially contributing to the reduced fibrin area observed in these patients. To address this point, we have revised the Discussion section (page 9, lines 6–10) to explicitly acknowledge this limitation and its possible impact on the interpretation of our findings:
“In our study, all patients undergoing rPCI received fibrinolytic therapy prior to thrombus aspiration. This treatment may have contributed to the lower fibrin content observed in the thrombi. Therefore, the potential effect of fibrinolytic therapy on thrombus composition should be carefully considered when interpreting the differences between the pPCI and rPCI groups.”
Reviewer Comment 2:
Table 2. It is hard to believe that the quantitative analysis of HE images and SEM images is accurate enough. In Table 2, both OM and SEM methods can detect the erythrocytes area. But the results from the two methods are dramatically different, and statistical results are also different. In my opinion, the conclusion is not convincing if it is method-dependent.
Response:
We understand and appreciate the reviewer’s concern. The OM and SEM analyses are complementary, as they examine different aspects of thrombus composition: OM assesses the thrombus core, while SEM focuses on the surface structure. This methodological distinction explains the differences observed between the two approaches in detecting erythrocytes.
To clarify this point, we have updated the Limitations section (page 11, lines 9–13) to emphasize these differences and discuss their implications for interpretation:
“OM and SEM target distinct areas of thrombus architecture. OM evaluates the core composition of the thrombus, while SEM predominantly examines surface features. These inherent methodological differences may contribute to variations in quantitative results and should be considered when interpreting the findings.”
Please also note that we stated in final sentence of the Limitations section:
“Finally, the present findings should be considered a hypothesis generated only, and further and larger studies are warranted.” (page 12, line 1)
Reviewer Comment 3:
Section 2.3. The HE images were analyzed using a semi-automated structural analysis based on color thresholds. What are the thresholds to segment different thrombus components? Please provide more details. If the authors used one threshold for all HE images, how did the authors address the color variation during the HE staining process?
Response:
We thank the reviewer for highlighting the need for clarification. The semi-automated structural analysis was performed using the cellSens Dimension V1 software (Olympus Corporation, Tokyo, Japan), which allowed segmentation of thrombus components based on histological characteristics in H&E-stained images.
We have added these details to the Methods section (page 8, lines 3–10):
“The slides were then stained with hematoxylin and eosin (H&E). For analysis, 3 to 4 images (depending on the thrombus size) were captured from random. Random areas were selected based on the sample quality, ensuring that the selected regions were free from artifacts (such as tissue folds and staining inconsistencies) and displayed clear histological features suitable for analysis. To achieve this, the slide was examined at low magnification (10×) to identify suitable regions, and then 3 to 4 fields (depending on the thrombus size) within those areas were randomly selected at 60× magnification for imaging. This method aimed to reduce bias by avoiding pre-selection based on specific histological features, while prioritizing regions with high-quality staining and structural integrity. The semi-automated structural analysis was performed using the cellSens Dimension V1 software to segment different thrombus components based on their histological characteristics in H&E-stained images. Standard color thresholds were defined to differentiate key thrombus components, including erythrocytes (green area), leukocytes (small red areas), and fibrin (yellow areas). These ranges were initially determined by analyzing representative images from each sample to establish baseline color profiles for each component. The thresholds were then applied consistently across all images within a sample to ensure uniformity in segmentation. Manual adjustments were performed when the observer noticed that the color segmentation did not adequately identify the corresponding structures in the original H&E staining.”
Unfortunately, it was not possible to retrieve specific numerical threshold values used for segmenting the thrombus components. To address this limitation, we have also noted in the Limitations section that:
Yet the semi-automated OM measurements could be subject to unintentional observer interference (page 2, lines 1-2).
Reviewer Comment 4:
Does all patients undergoing rPCI have a failed fibrinolytic therapy?
Response:
Yes, all patients undergoing rPCI in our study had failed fibrinolytic therapy. We have clarified this in the Methods section (page 4, lines 8–10):
“All patients in the rPCI group received Tenecteplase (30–50 mg bolus, adjusted for weight) and underwent rPCI as early as possible after documented fibrinolytic failure.”
Reviewer Comment 5:
Section 2.4. Each SEM image was analyzed with a minimum analyzed area of 5700 μm² per patient. What is the percent of the analyzed area (5700 μm²) of the entire sample area? If the percent is too small, could the analyzed area be a good/accurate representative of the whole sample?
Response:
The analyzed area of approximately 5700 μm² for each patient corresponds to roughly 50% of the total thrombus surface area, depending on the size of the aspirated material.
We have revised the Methods section (page 7, lines 3–5) as follows:
“This area corresponded to approximately 50% of the total thrombus surface. Although the analyzed area represents only a portion of the total thrombus surface, random field selection and systematic sampling were employed to enhance representativeness.”
Reviewer Comment 6:
Results Section. Please provide more details on why 5 thrombi only had OM, and 6 only had SEM? What are the sample sizes of these thrombi? And what is the minimal sample size eligible for OM or SEM in this study?
Response:
We thank the reviewer for this important question. In certain cases, the quantity of aspirated thrombus material was insufficient to divide for both OM and SEM processing. Additionally, some fragments allocated for OM or SEM could not be analyzed due to technical issues or limited material.
For cases in which material was available for only one type of analysis (OM or SEM), the selection criteria were consistent across all patients. For SEM, the method involved randomly selecting six images captured at 5000× magnification from an average of 30 images per sample. For OM, 3–4 images were captured from random fields depending on thrombus size.
Although SEM macro-images of many thrombi were available, measurements were not obtained for all of them. Likewise, OM measurements were not performed on the thrombi before histological preparation.
We have added these details to the Limitations section (page 11, lines 14–20):
“In certain cases, it was not feasible to obtain paired samples for both OM and SEM analyses, which may have introduced potential measurement bias. Moreover, since aspirated thrombi represent only a fraction of the thrombus involved in the atherothrombotic event, they may not comprehensively reflect the composition of the entire thrombus.”(page 12, lines 3-7).
Reviewer Comment 7:
Figure 2, redundant caption in the figure.
Response:
We thank the reviewer for noticing this redundancy. We have revised Figure 2 and removed the redundant caption.
Closing Remarks:
We hope that our revisions and clarifications adequately address the reviewers’ concerns. We are grateful for the insightful feedback, which has significantly improved the quality of our manuscript.
Sincerely,
Adriano Caixeta on behalf of all authors
Reviewer 2 Report
Comments and Suggestions for Authors
This is a good paper that provides important and valuable information on the differences in thrombus structure in patients who have experienced STEMI and subsequently undergone PCI, either as immediate (pPCI) or delayed (rPCI) intervention following thrombolytic therapy.
The authors demonstrate that thrombi in rPCI patients contain less fibrin, which is expected since fibrinolysis with tenecteplase should degrade fibrin fibers. However, the data (Table 3) obtained using optical and scanning electron microscopy suggest that fibrinolysis occurred primarily on the surface of the thrombus. This is an important observation for understanding the mechanisms and effectiveness of fibrinolytic therapy. It also raises the question of whether agents that can penetrate into the thrombus and break down its internal structure might be more effective—something tenecteplase alone cannot achieve.
A particular strength of the study is the comparison of findings obtained through light and electron microscopy, which provides a more comprehensive view of thrombus composition and yields complementary insights.
From a methodological standpoint, the study appears to be conducted carefully and cleanly. The original use of a morphometric grid adds credibility to the results.
Now, regarding the shortcomings:
The manuscript is written rather carelessly, resulting in numerous minor points that need to be corrected:
- In the Introduction (not just the Abstract), the abbreviation rPCI should be clearly defined.
- Figure 2 seems unnecessary; the diagonal scanning method could be described briefly in one or two sentences without the need for an illustration.
- In Figure 3, one of the captions uses a question mark (?) instead of the figure number (line 191).
- In Figure 4, leukocytes are labeled with a capital "L," whereas all other elements are labeled with lowercase letters. It would be better to use a consistent labeling style.
- The grouping in Figure 4 (rescue PCI: A to D and I to L; primary PCI: E to H) is somewhat confusing and makes the figure hard to interpret. It would be preferable to present the groups in two clearly separated blocks: one for rPCI and one for pPCI.
- Also in Figure 4, the brightness of the images is inconsistent—panels A, D, F, I, and K are significantly darker than the others. Adjusting brightness for consistency would improve readability.
- Throughout the text, the use of "vs" is confusing. In my opinion, when comparing rPCI and pPCI, it is logical to list rPCI first (i.e., rPCI vs pPCI). The manuscript often reverses this, which becomes particularly problematic in the Discussion, where it becomes unclear which data come from the authors and which from external sources (e.g., Silvain et al.). This should be standardized throughout the manuscript.
- In Table 1, the various symbols (crosses, section signs, asterisks) used to indicate statistical significance (e.g., p<0.05 or p>0.5) are not explained in the legend, making it difficult to interpret the data.
Finally, a fundamental question: Were SEM images obtained from cross-sections or only from the thrombus surface? The Methods section does not clarify this point, and Figure 4’s caption states “The panels display varying magnifications and sections of the thrombus.” If cross-sections were made, how were they consistently cut in such a way that only the cell surfaces were visible, with no internal structures? That seems technically very difficult. Table 3 supports the interpretation that only the surface was observed with SEM, as the results differ from those obtained using histological sections—something to be expected, given the known differences between surface and internal thrombus composition.
This issue should be clearly described in the Methods and addressed in the Discussion to help interpret the differences seen between optical and electron microscopy.
Once you have addressed these issues, I believe the article will be suitable for publication.
Author Response
Reply to Reviewers
Manuscript Title: Optical and Scanning Electron Microscopy Thrombus Findings in STEMI Patients Undergoing Primary Versus Rescue PCI
Manuscript ID: biomedicines-3717514
Journal: Biomedicines
Dear Editor and Reviewers,
We sincerely thank the Editor and reviewers for their thoughtful and constructive comments on our manuscript. We greatly appreciate the time and effort dedicated to reviewing our work and for the valuable suggestions that have helped us enhance the clarity and scientific rigor of the manuscript. Below, we provide a detailed, point-by-point response to each comment. All changes made in the revised manuscript are highlighted in yellow for easy reference.
Reviewer Comments and Authors’ Responses
Reviewer Comment 1: Clinical significance: One main result is that thrombi in patients with rPCI had reduced fibrin area compared to pPCI. Since the patients with rPCI had fibrinolytic therapy before harvesting the thrombus materials…
Response: We thank the reviewer for this important observation. Indeed, all patients in the rPCI group received fibrinolytic therapy (tenecteplase) before thrombus harvesting. We agree that fibrinolysis may have influenced thrombus composition, potentially contributing to the reduced fibrin area observed in these patients. To address this point, we have revised the Discussion section (page 9, second paragraph) to explicitly acknowledge this limitation and its possible impact on the interpretation of our findings:
“In our study, all patients undergoing rPCI received fibrinolytic therapy prior to thrombus aspiration. This treatment may have contributed to the lower fibrin content observed in the thrombi. Therefore, the potential effect of fibrinolytic therapy on thrombus composition should be carefully considered when interpreting the differences between the pPCI and rPCI groups.”
Reviewer Comment 2: Table 2. It is hard to believe that the quantitative analysis of HE images and SEM images is accurate enough. In Table 2, both OM and SEM methods can detect the erythrocytes area. But the results from the two methods are dramatically different, and statistical results are also different. In my opinion, the conclusion is not convincing if it is method-dependent.
Response: We understand and appreciate the reviewer’s concern. The OM and SEM analyses are complementary, as they examine different aspects of thrombus composition: OM assesses the thrombus core, while SEM focuses on the surface structure. This methodological distinction explains the differences observed between the two approaches in detecting erythrocytes.
To clarify this point, we have updated the Limitations section (page 11, last paragraph) to emphasize these differences and discuss their implications for interpretation:
“OM and SEM target distinct areas of thrombus architecture. OM evaluates the core composition of the thrombus, while SEM predominantly examines surface features. These inherent methodological differences may contribute to variations in quantitative results and should be considered when interpreting the findings.”
Please also note that we stated in final sentence of the Limitations section:
“Finally, the present findings should be considered a hypothesis generated only, and further and larger studies are warranted.” (page 12, first paragraph)
Reviewer Comment 3: Section 2.3. The HE images were analyzed using a semi-automated structural analysis based on color thresholds. What are the thresholds to segment different thrombus components? Please provide more details. If the authors used one threshold for all HE images, how did the authors address the color variation during the HE staining process?
Response: We thank the reviewer for highlighting the need for clarification. The semi-automated structural analysis was performed using the cellSens Dimension V1 software (Olympus Corporation, Tokyo, Japan), which allowed segmentation of thrombus components based on histological characteristics in H&E-stained images.
We have added these details to the Methods section (page 8, lines 3–10):
“Each sample was fixed in 10% buffered formalin for 24 hours and then embedded in paraffin. Up to four sequential 5-μm-thick histological sections were cut using a rotary microtome and stained with hematoxylin and eosin (H&E). For analysis, three to four images (depending on thrombus size) were captured from randomly selected fields. To minimize bias and ensure quality, slides were first examined at 10× magnification to identify artifact-free regions with clear histological features (e.g., free from tissue folds or staining inconsistencies). These fields were then randomly selected within those regions at 60× magnification for imaging. Semi-automated structural analysis was performed using cellSens Dimension V1 software (Olympus Corporation, Tokyo, Japan), which segmented objects by color thresholding based on H&E staining (Figure 1). Red blood cells, white blood cells, and fibrin were identified as distinct, non-connected objects. Threshold values were adjusted to optimize pixel selection according to staining intensity. Measurement parameters, including object class, count, and summed area per component, were displayed in the results sheet, enabling direct comparison of components by their assigned colors. Manual adjustments were made when color segmentation did not accurately represent structures in the original H&E images.”
Unfortunately, it was not possible to retrieve specific numerical threshold values used for segmenting the thrombus components. To address this limitation, we have also noted in the Limitations section that:
Yet the semi-automated OM measurements could be subject to unintentional observer interference (page 2, lines 1-2).
Please see below the figure that explain the methodoly we used.
Reviewer Comment 4: Does all patients undergoing rPCI have a failed fibrinolytic therapy?
Response: Yes, all patients undergoing rPCI in our study had failed fibrinolytic therapy. We have clarified this in the Methods section (page 4, lines 8–10):
“All patients in the rPCI group received Tenecteplase (30–50 mg bolus, adjusted for weight) and underwent rPCI as early as possible after documented fibrinolytic failure.”
Reviewer Comment 5: Section 2.4. Each SEM image was analyzed with a minimum analyzed area of 5700 μm² per patient. What is the percent of the analyzed area (5700 μm²) of the entire sample area? If the percent is too small, could the analyzed area be a good/accurate representative of the whole sample?
Response: The analyzed area of approximately 5700 μm² for each patient corresponds to roughly 50% of the total thrombus surface area, depending on the size of the aspirated material.
We have revised the Methods section (page 7, lines 3–5) as follows:
“This area corresponded to approximately 50% of the total thrombus surface. Although the analyzed area represents only a portion of the total thrombus surface, random field selection and systematic sampling were employed to enhance representativeness.”
Reviewer Comment 6: Results Section. Please provide more details on why 5 thrombi only had OM, and 6 only had SEM? What are the sample sizes of these thrombi? And what is the minimal sample size eligible for OM or SEM in this study?
Response: We thank the reviewer for this important question. In certain cases, the quantity of aspirated thrombus material was insufficient to divide for both OM and SEM processing. Additionally, some fragments allocated for OM or SEM could not be analyzed due to technical issues or limited material.For cases in which material was available for only one type of analysis (OM or SEM), the selection criteria were consistent across all patients. For SEM, the method involved randomly selecting six images captured at 5000× magnification from an average of 30 images per sample. For OM, 3–4 images were captured from random fields depending on thrombus size. Although SEM macro-images of many thrombi were available, measurements were not obtained for all of them. Likewise, OM measurements were not performed on the thrombi before histological preparation.
We have added these details to the Limitations section (page 11, lines 14–20):
“In certain cases, it was not feasible to obtain paired samples for both OM and SEM analyses, which may have introduced potential measurement bias. Moreover, since aspirated thrombi represent only a fraction of the thrombus involved in the atherothrombotic event, they may not comprehensively reflect the composition of the entire thrombus.”(page 12, lines 3-7).
Reviewer Comment 7: Figure 2, redundant caption in the figure.
Response: We thank the reviewer for noticing this redundancy. We have revised Figure 2 and removed the redundant caption.
Reply to Reviewer 2.
Reviewer Comments and Authors’ Responses
Reviewer Comment 1: In the Introduction (not just the Abstract), the abbreviation rPCI should be clearly defined.
Response. We thank the reviewer for this observation. We have revised the Introduction section (page 3, lines 26) to ensure that “rescue PCI (rPCI)” is clearly defined at its first occurrence, as well as “primary PCI (pPCI).”
Reviewer Comment 2: Figure 2 seems unnecessary; the diagonal scanning method could be described briefly in one or two sentences without the need for an illustration.
Response: We have removed Figure 2 from the manuscript and included a concise description of the diagonal scanning method directly in the Methods section (pages 6-7, Section: Scanning Electron Microscopy (SEM). The new updated paragraph states:
“A 190 µm² grid with a random offset was then applied to each selected image. Morphometric grid-based methods typically aim for the grid cell size to accommodate the visualization of 3 to 4 units of the studied structure. Consequently, the grid size was determined based on the average dimensions of leukocytes and erythrocytes. This parameter of a 190 µm² grid was consistently applied to other structures—fibrin, platelets, and cholesterol crystals—to ensure uniformity in the appraisal process. Using the same grid and offset for all structures enhanced the representativeness of the material's composition and organization. Since these analyses are highly labor-intensive, area measurements were conducted in alternating full grid spaces, starting with the first complete square at the top left of the image.”
Reviewer Comment 3: In Figure 3, one of the captions uses a question mark (?) instead of the figure number (line 191).
Response: We thank the reviewer for noticing this error. The caption for Figure 3 has been corrected to include the appropriate figure number.
Reviewer Comment 4: In Figure 4, leukocytes are labeled with a capital “L,” whereas all other elements are labeled with lowercase letters. It would be better to use a consistent labeling style.
Response: The labels in Figure 4 have been revised to ensure consistent use of lowercase letters across all elements.
Reviewer Comment 5: The grouping in Figure 4 (rescue PCI: A to D and I to L; primary PCI: E to H) is somewhat confusing and makes the figure hard to interpret. It would be preferable to present the groups in two clearly separated blocks: one for rPCI and one for pPCI.
Response: To enhance clarity, Figure 4 has been reorganized (now Figure 3 in this updated version) for better presentation of the rPCI and pPCI images in two separate blocks. In addition, select images in the two blocks have been removed so that all studied elements are represented (with rPCI above and pPCI below). The caption has also been updated accordingly.
Reviewer Comment 6: Also in Figure 4, the brightness of the images is inconsistent—panels A, D, F, I, and K are significantly darker than the others. Adjusting brightness for consistency would improve readability.
Response: The brightness and contrast of all panels in Figure 4 (now Figure 3 in this updated version) have been adjusted to ensure visual consistency and improve readability.
Reviewer Comment 7: Throughout the text, the use of “vs” is confusing. In my opinion, when comparing rPCI and pPCI, it is logical to list rPCI first (i.e., rPCI vs pPCI). The manuscript often reverses this, which becomes particularly problematic in the Discussion, where it becomes unclear which data come from the authors and which from external sources (e.g., Silvain et al.). This should be standardized throughout the manuscript.
Response: We have carefully reviewed the entire manuscript and standardized the use of “rPCI vs pPCI” throughout, ensuring consistent ordering in all sections, including the Discussion. For instance:
In the Abstract we now write in the Methods:
“Patients who underwent rPCI had significantly higher C-reactive protein levels and a longer ischemic interval at admission compared to those treated with pPCI (9.92 hours [range: 1.58–106.17] vs. 2.14 hours [range: 0–48]; p < 0.001). Optical microscopy analysis revealed that thrombi from rPCI patients exhibited a significantly higher erythrocyte area percentage (18.36% [range: 0.3–50.08] vs. 0.91% [range: 0–70.1]; p = 0.001), a lower fibrin content as assessed by optical microscopy (79.49% [range: 49.2–98.25] vs. 94.43% [range: 29.19–99.92]; p = 0.006), and a greater amount of cholesterol crystals as measured by SEM (1.73 μm² [range: 0–18.51] vs. 0.08 μm² [range: 0–0.71]; p < 0.001).”
In the Results we now state:
“The data were then statistically analyzed to compare thrombus composition between the rPCI and pPCI groups. Compared to patients undergoing pPCI, those in the rPCI group exhibited thrombi with a significantly higher erythrocyte area percentage (18.36% [range: 0.3–50.08] vs. 0.91% [range: 0–70.1]; p = 0.001) and a lower fibrin content (79.49% [range: 49.2–98.25] vs. 94.43% [range: 29.19–99.92]; p = 0.006), as assessed by optical microscopy (Figure 2). Scanning electron microscopy revealed a significantly greater amount of cholesterol crystals in the rPCI group (1.73 μm² [range: 0–18.51] vs. 0.08 μm² [range: 0–0.71]; p < 0.001) (Table 2). Additionally, there was a trend toward a higher leukocyte area in the rPCI group based on SEM analysis (1.44 μm² [range: 0.03–20.65] vs. 0.54 μm² [range: 0–21.88]; p = 0.07) (Figure 3).”
Reviewer Comment 8: In Table 1, the various symbols (crosses, section signs, asterisks) used to indicate statistical significance (e.g., p<0.05 or p>0.5) are not explained in the legend, making it difficult to interpret the data.
Response: We thank the reviewer for this observation. The legend for Table 1 has been revised to remove all symbols. Originally, the symbols were included to indicate which statistical test had been applied to each specific column. However, to simplify the table—and since the statistical methods are already described in the Statistical Analysis section—removing the symbols does not compromise the reader’s understanding of Table 1.
Reviewer Comment 9: A fundamental question: Were SEM images obtained from cross-sections or only from the thrombus surface? The Methods section does not clarify this point, and Figure 4’s caption states “The panels display varying magnifications and sections of the thrombus.” If cross-sections were made, how were they consistently cut in such a way that only the cell surfaces were visible, with no internal structures? That seems technically very difficult. Table 3 supports the interpretation that only the surface was observed with SEM, as the results differ from those obtained using histological sections—something to be expected, given the known differences between surface and internal thrombus composition. This issue should be clearly described in the Methods and addressed in the Discussion to help interpret the differences seen between optical and electron microscopy.
Response: We thank the reviewer for bringing this critical point to our attention. Indeed, all SEM images were obtained from the surface of the thrombi rather than cross-sections. We have clarified this in the Limitations section (page 11, last paragraph). In addition, please refer to the response to Reviewer 2's question, which addresses the same issue; we have updated the Limitations section (page 11, last paragraph) to emphasize these differences and discuss their implications for interpretation.
“Third, OM and SEM target distinct areas of thrombus architecture. OM evaluates the core composition of the thrombus, while SEM predominantly examines surface features. “SEM analysis was performed exclusively on the surface of the thrombus, as cross-sectional imaging was not feasible with the applied preparation techniques. These inherent methodological differences may contribute to variations in quantitative results and should be considered when interpreting the findings.”
Closing Remarks:
We hope that our revisions and clarifications adequately address the reviewers’ concerns. We are grateful for the insightful feedback, which has significantly improved the quality of our manuscript.
Sincerely,
Adriano Caixeta on behalf of all authors
Round 2
Reviewer 1 Report
Comments and Suggestions for Authors
My comments are well addressed.